# A Review of Noninvasive Methodologies to Estimate the Blood Pressure Waveform

**DOI:** 10.3390/s22103953

**Published:** 2022-05-23

**Authors:** Tasbiraha Athaya, Sunwoong Choi

**Affiliations:** School of Electrical Engineering, Kookmin University, Seoul 02707, Korea; athayatasbiraha@kookmin.ac.kr

**Keywords:** blood pressure waveform, noninvasive, deep learning, ultrasound, pressure

## Abstract

Accurate estimation of blood pressure (BP) waveforms is critical for ensuring the safety and proper care of patients in intensive care units (ICUs) and for intraoperative hemodynamic monitoring. Normal cuff-based BP measurements can only provide systolic blood pressure (SBP) and diastolic blood pressure (DBP). Alternatively, the BP waveform can be used to estimate a variety of other physiological parameters and provides additional information about the patient’s health. As a result, various techniques are being proposed for accurately estimating the BP waveforms. The purpose of this review is to summarize the current state of knowledge regarding the BP waveform, three methodologies (pressure-based, ultrasound-based, and deep-learning-based) used in noninvasive BP waveform estimation research and the feasibility of employing these strategies at home as well as in ICUs. Additionally, this article will discuss the physical concepts underlying both invasive and noninvasive BP waveform measurements. We will review historical BP waveform measurements, standard clinical procedures, and more recent innovations in noninvasive BP waveform monitoring. Although the technique has not been validated, it is expected that precise, noninvasive BP waveform estimation will be available in the near future due to its enormous potential.

## 1. Introduction

For more than 50 years, indwelling arterial catheterization has enabled the bedside measurement of continuous arterial pressure values using waveform analysis [1]. According to a World Health Organization (WHO) report, hypertension, more commonly referred to as high blood pressure (BP), is a global public health problem. Globally, this disease affects billions of people, with two-thirds of them living in middle- to low-income countries [2]. Additionally, hypertension increases the risk of developing cardiovascular disease [3], kidney disease [4], heart attack [5], and diabetes [6,7]. High BP was responsible for nearly 45% of deaths due to heart disease, and stroke was the reason for 51% of the deaths [2,8]. Furthermore, regulating this medical condition is difficult and expensive [9].

Invasive blood pressure waveform monitoring has been used in critical patients in both ICUs and operating rooms to aid in the rapid diagnosis of cardiovascular insufficiency and the monitoring of response to medications used to correct irregularities prior to the onset of hypotension or hypertension. BP waveform variation provides a wealth of information about an individual’s dynamic cardiovascular state [10,11]. The unique architecture of the venous BP waveform is intimately related to significant right cardiac activities, whereas each of the peaks and valleys in the arterial blood pressure (ABP) waveform represents a distinct left heart function [12,13]. As a result, interest in BP waveform analysis has increased significantly. Numerous critical parameters can be determined through the use of BP waveform analysis. Among them are the prediction of vascular resistance, left ventricular stroke volume (SV), variation of SV, and pulse pressure during positive-pressure respiration [14]. Real-time monitoring of BP variations is possible with an arterial catheter equipped with a pressure transducer. Additionally, it enables earlier detection of intraoperative hypotension than indirect measurement techniques do. It also provides reliable venous access for blood sampling. Invasive BP waveform monitoring enables pressure monitoring in situations when noninvasive BP monitoring with a cuff is not possible. One such case is during cardiopulmonary bypass when nonpulsatile blood flow occurs continuously [15,16].

According to the history of BP measurement, Stephen Hales was the first person to measure invasive BP in 1733. He took BP readings from a horse’s carotid and femoral arteries. He measured BP by inserting a brass tube connected to a vertical glass tube into the horse’s trachea [17,18]. Following that, the method was refined by additional physicians and scientists. In 1828, Jean-Louis Poiseuille and Daniel Bernoulli [19] used a U-shaped pipe filled with mercury to measure pressure at various locations along the arteries [20]. In 1949, Peterson and his colleagues implanted the first therapeutically useful artery catheter. When a small plastic catheter is inserted into an artery using a needle and then removed, the catheter remains inside the artery. The capacitance manometer used in this technique enables long-term recording of the BP waveform without discomfort and allows for relatively free movement of the subject [21]. Peirce [22] and Seldinger [23] have since explained numerous techniques. Seldinger pioneered the “catheter-over-wire” technique which has become a widely used technique in recent years. By 1990, over eight million invasive arterial catheterization probes had been implanted. Monitoring of the invasive BP waveform is crucial in situations where a wide variation in BP is observed, precise BP regulation is required for the treatment of end-organ disease, blood gas measurements are required frequently or multiple times, SV and cardiac output must be monitored continuously, and patients have critical needs (e.g., dysrhythmias, trauma, or burn patients), among others [24].

Although invasive BP waveform estimation is the gold standard for BP waveform monitoring as well as measurements of SBP, DBP, and mean arterial pressure (MAP) values due to its excellent accuracy, it is too intrusive for routine inspections due to patient discomfort, increased infection risk, prolonged cannulation, hematoma formation, catheter embolism, thrombosis with distal ischemia, arterial drug administration, blood loss, vasospasm, multiple insertion attempts, nerve or adjacent body structure damage, high-dose vasopressor administration, etc. [10,25,26].

Given the risks associated with invasive BP waveform estimation, extensive research has been conducted on noninvasive BP estimation. The majority of noninvasive techniques attempt to strike a clinical balance between the arterial catheter and cuff-based techniques [27,28,29,30,31]. Currently, there are only a few studies that have attempted to establish methodologies for noninvasive BP waveform prediction. Vascular unloading techniques based on optical methods [32] and applanation tonometry [33] methods have the potential to estimate the BP waveforms. However, these methods face numerous technical challenges. A wearable ultrasonic probe has been proposed to circumvent these limitations [34,35,36]. However, the method has introduced some new difficulties. The volume of medical data available in the electronic health records enables machine learning or deep learning algorithms with physiological signals such as photoplethysmogram (PPG) or electrocardiogram (ECG) or both to estimate the BP waveform in a more convenient way [37,38,39,40,41]. However, none of the proposed methods has been validated.

The purpose of this article is to summarize all available methods for noninvasive BP waveform estimation. Our main purpose is to review the studies that predicted the BP waveforms, and then we show the results of BP values (SBP, DBP, MAP) that are obtained from the estimated noninvasive BP waveforms. Noninvasive BP monitoring contains a vast area of research. We specifically searched for the related papers that predicted blood pressure waveform or arterial blood pressure waveform noninvasively. Recently, research on noninvasive BP waveform monitoring has been gaining popularity. The following is the structure of the paper: To begin, a summary of the existing invasive and noninvasive approaches used in BP waveform estimation studies is provided. Following that, all available noninvasive BP waveform estimation methods are summarized. This is accomplished through a discussion of the studies that have been conducted thus far, as well as their relative merits and demerits. The review article concludes with some recommendations for future research in this field.

## 2. Searching Strategy

This systematic review article was written using the Google Scholar and Web of Science databases. As several articles reviewed different techniques to predict noninvasive BP values (SBP, DBP, and MAP), we focused on the studies that predicted the full waveforms and used the keywords that can provide studies specific to BP waveform prediction [42,43,44,45,46,47]. The web search was restricted to journal and conference articles published until January 2022. The search was conducted using the following keywords:Hypertension or high blood pressure;Arterial waveform;Blood pressure waveform;Machine learning in ABP waveform;Signal processing in ABP waveform.

Each article was classified using the data, technique, and algorithms described in it. Additionally, this search technique considered the variables in the dataset and their relationship to the BP waveform as well as the data processing methods used to assess physiological data. There are several review articles available on the Web of Science and Google Scholar that discuss the methodologies for estimating SBP, DBP, and MAP values without using a cuff [42,43,44,45,46,47]. Some studies mentioned two techniques (applanation tonometry and the volume clamp method) along with the cuffless methods [48,49,50,51,52,53]. However, these two methods are discussed to review the old techniques, and the main purpose of the reviews is to review the papers that can measure SBP, DBP, and MAP values rather than focusing to estimate the complete BP waveforms. It is worth mentioning that applanation tonometry and the volume clamp method have been studied for a long time (since 1963 [54]) to replace invasive BP waveform, and still there are studies being conducted. However, the processes are not yet established. Moreover, the pressure-based method has several disadvantages. To address the limitations, in recent time new methods using ultrasound and deep learning have been developed to predict BP waveform noninvasively. No other review paper has reviewed recent methods to particularly predict BP waveforms other than old studies related to applanation tonometry and the volume clamp methods. To our knowledge, this is the first paper to review all the available noninvasive methods for estimating BP waveforms.

## 3. Noninvasive BP Waveform Estimation Methods

After extensive research, we discovered three distinct methods for estimating noninvasive BP waveforms.

### 3.1. Pressure-Based Method

#### 3.1.1. Vascular Unloading Technique

The vascular unloading technique or volume clamp method is a noninvasive blood pressure waveform measurement technique based on a modified Peáz principle [55]. The outside pressure from the finger cuff linearizes the pressure inside the artery, maintaining a constant blood volume. On the index or middle finger, a cuff is placed along with an infrared (IR) light source and a photodiode inside the cuff. It needs to be mentioned that multiple concentrically interlocking loops were introduced [32] following the failure of the single loop system [55,56,57,58,59,60,61,62,63,64]. In the multiple loop system, fast adjustments are performed by controlling the inside loops. Fast adjustments mean fast pressure building and release, controlling pressure, light and fast unloading, continuous change in BP, and surrounding a light filter. The inside loops also provide almost ideal constraints for the loops controlling the outside. The outside loops are required for providing the system with long-term stability.

The light source (LED) and the photodiode are used to determine the volume of blood in the arteries. Consistency of the generated PPG signal can be maintained by controlling the variable cuff pressure. When the blood volume in the finger increases during systole, the control system illustrated in Figure 1 increases the cuff pressure in the finger until the excess blood volume is squashed out. During diastole, the blood volume within the finger decreases. As a result, the cuff pressure decreases, while the finger’s total blood volume remains unchanged. Due to the fact that both the PPG signal and the blood volume remain constant over time, the intra-arterial pressure equals the finger cuff pressure. This pressure is determined using a manometer, and a BP waveform is generated using the continuous manometer reading. The advantages and disadvantages of the pressure-based vascular unloading technique are stated in Table 1.

#### 3.1.2. Arterial Tonometry

Pressman and Newgard introduced applanation tonometry (AT) in 1963, drawing inspiration from Vierordt and Marey’s pioneering work as well as previous ocular monitoring studies [65]. Tonometry refers to the measurement of pressure, while applanation refers to flattening. A hand-held strain gauge pressure sensor, referred to as a tonometer, is placed on the radial artery, and light pressure is applied to slightly flatten the artery until it begins to deform as shown in Figure 2. A piezoelectric pressure sensor is generally used to monitor BP waveforms in the radial artery of the wrist [66]. The measurement site is chosen to allow for the presence of a bony structure beneath the artery [67,68]. The vertical displacements observed by the tonometer at this point are proportional to the artery pressure when the measuring apparatus is modeled as a linear springy model [54]. Finally, by maintaining consistent positioning of the system, a strain gauge sensor converts the measured vertical displacements into electrical signals representing ABP waveforms [33,65].

The advantages and disadvantages of the pressure-based arterial tonometry method are stated in Table 2.

### 3.2. Ultrasound-Based Method

The ultrasound-based approach describes the design and working process of an ultrasonic probe that is attached to the skin and capable of recording BP waveforms in deeply embedded veins and arteries [34]. This method addresses the limitation of the ultrasound method that uses an imaging probe [35,36]. As mentioned in [34], the wearable ultrasonic probe is 240 μm thin and can be stretched up to 60% with strains. The process of measuring BP waveforms using the ultrasound device is depicted in Figure 3. The probe continuously measures the diameter of a palpitating blood vessel. Using mathematical equations, the continuously measured diameter is converted to BP waveforms [69]. The authors assumed that the human blood vessel is elastic and has very little viscoelasticity; so the BP waveform can be calculated from the vessel diameter waveforms using Equation (1).
(1)$$pr\left(t\right)=pr_{d}\times e^{\alpha \left(\frac{ar\left(t\right)}{ar_{d}}-1\right)}$$
where $pr_{d}$ is the DBP acquired by wearing a commercial BP cuff on the brachial artery, the cross-section of the arterial diastole is denoted as $ar_{d}$, and the coefficient of vessel rigidity is α. The artery is assumed to be rotationally symmetrical, and $ar\left(t\right)$ is calculated using Equation (2).
(2)$$ar\left(t\right)=\frac{\pi D^{2}\left(t\right)}{4}$$
where $D\left(t\right)$ is the diameter waveform of the artery, which is measured using the ultrasound wearable device. Then, α can be calculated by Equation (3).
(3)α=ard×lnprsprdars−ard
where $ar_{s}$ is the cross-section of the arterial systole, and the SBP measured using the same BP cuff is denoted as $pr_{s}$. The authors used the aforementioned equations and a calibration process for α and $pr_{d}$ to estimate the BP waveform $pr\left(t\right)$.

The ultrasound probe’s transducer is activated during each pulse cycle with a 7.5 MHz frequency alternating current (AC) to obtain the ultrasound waveforms. As illustrated in Figure 3, ultrasound wall tracking was used to determine the temporal resolution. This was accomplished by generating ultrasound waveforms with a high pulse repetitive frequency (PRF). This PRF determines the number of pulse cycles per second. When the generated ultrasound waveforms come into contact with various biological interferences, some of the waveforms are reflected, and others are transmitted through the interferences. The reflections are detected using the same transducer that generates the ultrasound waveforms. When the acoustic velocity of the tissue is known, the reflected waveform (i.e., time of flight) contains information about the biological interference locations. The spatial pulse length (SPL) is sufficiently long to distinguish between the posterior and anterior walls of blood vessels, as is customary when measuring vascular diameter, as illustrated in Figure 3. The advantages and disadvantages of the ultrasound-based method are listed in Table 3.

### 3.3. Deep-Learning-Based Methods

Deep learning algorithms trained on biomedical signals have gained considerable popularity in recent years in the field of BP waveform estimation due to their ability to automatically learn critical features. With the abundance of biomedical data available, deep-learning-based techniques for BP waveform prediction have become a highly researched topic in recent years.

During our review process, we discovered that the majority of papers used a variety of deep learning algorithms to predict BP waveforms using biosignals such as photoplethysmogram (PPG) and electrocardiogram (ECG). Arterial BP (ABP) waveforms are used as the gold standard and reference values in these works. These three waveforms are depicted in Figure 4. Through the use of body patches, an ECG is obtained from the heart. ABP waveforms are typically acquired via arterial catheter from the radial artery. PPG signals can be generated by a photodiode embedded in the finger. ABP signals can also be acquired from the brachial arteries. Additionally, the PPG signal has multiple acquisition points (e.g., wrist, toe, earlobe, etc.). The waveform shapes vary according to the measurement locations. PPG and fingertip ABP have an almost identical shape, as illustrated in Figure 4.

Due to the structural similarity, the majority of the papers discussed in Table 4 used PPG signals to estimate BP waveforms. In general, Figure 5 illustrates the steps involved in deep-learning-based BP waveform estimation using biosignals. Each of the works discussed thus far as summarized in Table 4 demonstrates a process that is more or less similar.

Table 4 summarizes the methods, input signals used, and length of the input signals, along with the publication year. Table 5 summarizes the advantages and disadvantages of deep-learning-based methods.

Ref. [70] proposes an optimized wavelet neural network with PPG signals as input for noninvasively predicting BP waveforms. According to the authors, implementing the proposed structure is significantly easier than implementing a three-layered wavelet neural network because the structure of cache hidden layer nodes without multipliers is significantly simpler than the structure of wavelet hidden layer nodes. Additionally, the paper proposed an algorithm dubbed inhomogeneous resilient backpropagation to reduce computational complexity and speed of convergence (IRBP). The algorithm determines the hidden nodes’ weights. However, as mentioned in Table 5, this model has high computational complexity and also suffers from data redundancy [71].

The authors of [72] employed a Long Short-Term Memory (LSTM) recurrent neural network as the proposed model’s input. The network architecture and overall process, on the other hand, are not described in detail. They trained the model individually for each patient, indicating that the model is not generalizable. When multiple inputs are used, the output is a single ABP point. The training process is lengthy and inefficient for implementing a device.

The work in [41] proposed an ANN model and dubbed it the nonlinear autoregressive model with exogenous input (NARX). They predicted BP waveforms using electrocardiography (ECG) signals. They later published an expanded version of their paper in [73] as a journal. In [73], they used the same model but demonstrated that it can be used with either PPG or ECG signals or both. When both signals were used, the highest Pearson correlation coefficient was obtained. However, two sets of BP data are required for model training. The delay removal procedure is not appropriate in all circumstances. During training, the ECG, PPG, and BP peak ranges were not unified. Cross-correlation analysis was used to quantify any difference in predicted and measured blood pressure.

PPG2ABP is the name given to their work in [37], which is available on the preprint server. They estimated the BP waveform using two deep learning models in this work. To begin, they estimated the waveform using an approximation network, which is a one-dimensional U-Net network fed with a PPG signal. They then corrected the estimated BP waveforms using a refinement network. They refined the model using a 1D MultiResUNet model. Another study [38] proposed estimating BP waveforms using only a 1D modified U-Net network. The two papers differ primarily in their signal preprocessing techniques. Another publication [74], which is also available on the preprint server, used U-Net to predict the BP waveform using a different preprocessing method. Ref. [74] demonstrated the feasibility of implementing their proposed model on a Raspberry Pi 4 device with an inference time of 4.25 ms. However, the implementation process was not detailed. As the U-Net model requires high computational complexity due to a large number of parameters, device implementation is difficult with this model. They did, however, include an average ensemble block prior to the encoder and a denoising block following the decoder. The average ensemble block aids in model convergence during training, while the denoising block denoises the output signal to produce a less-distorted BP waveform.

In [40], a comparison of two deep convolution autoencoders named LeNet-5 and U-Net to estimate the BP waveforms is shown. To investigate data generalization, the cross–validation (CV) technique was used. The results indicate that the U-Net outperforms other estimation methods for SBP values. Meanwhile, the LeNet-5 is marginally more accurate at predicting DBP values. Finally, a genetic algorithm-based optimization deep convolutional autoencoder (GDCAE) is used to optimize the ensemble of CV models. According to the findings, the GDCAE outperforms both the LeNet-5 and the U-Net. Thus, this review discusses the outcome of the best-performing model GDCAE. However, combining two deep learning algorithms to obtain two distinct values requires a large number of parameters, which is inefficient computationally. Additionally, two values can be obtained by combining two different models, but no optimized model for predicting BP waveforms is shown.

Additionally, the authors of [39] also used the PPG signal to predict the BP waveforms. They proposed a deep autoencoder based on regularized convolution, abbreviated as RDAE, for this purpose. They provided two versions of the model: one that is RDAE-based and another that is RDAE-based with calibration. They estimated the BP waveform and then used the waveforms to predict the SBP, DBP, and MAP. They demonstrated that the proposed model requires fewer parameters than alternative methods.

A 1D V-Net deep learning algorithm is proposed for BP waveform prediction in [25]. Two signals (ECG and PPG), with a 4 s window each, were used as input to the model, along with several constant values. Constants were encoded and treated as additional channels at each timestep. The following constants were used: the most recent noninvasive SBP, DBP, and MAP values obtained prior to the window, the time interval between these measurements, the standard deviation (STD) and median of the pulse arrival time, and the pulse rate. There is a residual difference between the PPG and BP waveforms at the input. The model was constructed in such a way that it is capable of learning the residual error. The primary issue is that the model requires a large number of input variables and also requires noninvasive blood pressure measurements.

Another recent work [75] published on the preprint server, proposed the use of a cycle generative adversarial network (CycleGAN) to predict BP waveforms using PPG data as input. Despite the fact that the majority of the papers used an encoder–decoder method, they used a generator and a discriminator network to estimate the waveforms, which is a novel technique. They conducted training and testing using data from the same subject as described in Table 5.

#### 3.3.1. Data Preprocessing

Preprocessing data is critical for deep learning methods to provide an accurate model estimation. Even when similar algorithms and data are used, subtle differences in preprocessing techniques result in noticeable differences in the results as shown in Figure 5. Typically, data preprocessing entails one or more of the following:Segmenting the data to train the model;Removing erroneous biosignals that are inaccurate for measurement;Filtering the biosignals to remove the baseline wandering and high–frequency noises;Normalizing inputs and outputs for accurate training of the model.

Biosignals contain a variety of artifacts. If those artifacts are used to train deep learning algorithms, the algorithm may produce incorrect results. As a result, the erroneous data containing artifacts must be deleted. As all of the papers listed in Table 4 used either PPG or ECG signals or a combination of the two, signal filtering is necessary to remove high-frequency noise and extract the necessary portion of the signal for the algorithm. Normalization of the data is an additional step in data preprocessing. To eliminate the range difference between different biosignals, the signals must be normalized. This is accomplished through the use of a variety of different normalization techniques. Table 6 illustrates the differences in the data preprocessing methods used by various works.

Ref. [70] uses the PPG signal as an input and the ABP signal as an output. The input’s length is not specified. Due to the fact that the energy in PPG signals is primarily in the low–frequency range, below 20 Hz, this work makes use of the low-pass filter for the Daubechies wavelet. Scales 21 and 22 were chosen to represent the frequency range over which the PPG signal will be processed. This article makes no use of any other preprocessing technique. Ref. [72] eliminated baseline drift from both PPG and ABP signals by removing the output of the linear square fit. The input size is equal to the proposed network’s node count.

Another work [73] used double derivation to determine the signal’s erroneous portion. They established a cutoff point of ±5. The signal segments with the highest standard derivations of the threshold value were excluded from the dataset. To locate noisy data for some signals, up to six standard derivations were used. The input sample size was 100 to obtain one BP point.

In [37], signal filtering was accomplished using a simple averaging filter. Signals with DBP values less than 50 mmHg and SBP values greater than 200 mmHg were considered to have irregular BP values and were excluded from the dataset. Then signals with unacceptably fast heartbeats and long discontinuities were eliminated as well. A PPG signal with a duration of 8.192 s (1024 samples) was used as the input, and a signal with a similar size was obtained as the output. By setting the coefficients for decomposition to zero, Daubechies wavelet denoising, as described in [70], was used to neglect too low and too high-frequency components by setting the coefficients for decomposition to zero value. Finally, the authors used mean normalization for the PPG signal. In [40], each input PPG and output ABP were 5 s in length. No other filtering process was used to identify erroneous segments; they were identified manually through visual inspection.

The method of preprocessing is described in detail in [38]. To begin, they used a bandpass Equiripple FIR filter with a frequency range of 0.5–8 Hz to filter PPG data. The PPG and ABP signals were then segmented into 350 samples with a 100-sample overlap. Following that, a machine learning model was used to identify and remove erroneous signals from the dataset. Signals were measured at distinct locations. Thus, a phase difference exists between the two signals. By using cross-correlation, the ABP signals were phase-matched to the PPG signals. Following that, 256 samples of phase-matched signals were chosen to train, validate, and test the proposed 1D U-Net model. The duration of the input and output signals was 2.048 s.

The work [39] clarified their preprocessing technique. They sequentially performed signal filtering, excluding erroneous segments, segmentation, and normalization. They used distinct procedures to filter both PPG and ABP signals. PPG signals were filtered with a fourth order bandpass filter and out-of-range peaks and valleys were clipped. They used the Savitzky–Golay filter to filter the ABP signals. They then ensured that the model had an independent identity distribution for training, validation, and testing. They classified the ABP signals into three categories based on the SBP and DBP values: normal, prehypertensive, and hypertensive. Then, in a 6:2:2 ratio, they divided the signals into three classes and created the final dataset. Next, they excluded erroneous ABP signals from the final dataset when any abnormal condition was found. After that, the signals were segmented into 625 samples to use for the input and output. Both the input and output segments are Z-score normalized.

Ref. [74] defines ABP signals with 60 mmHg ≥ DBP ≥ 130 mmHg and 80 mmHg ≥ SBP ≥ 180 mmHg as irregular signals and excludes them from the dataset. Then, similar to [70], a preprocessing technique is used. One additional step is that the work downsampled the data to satisfy the computational requirement.

The article [25] employed a thorough preprocessing technique. Along with the PPG and ECG signals, this paper required several input parameters. All signals were downsampled to a 100 Hz frequency. The signals were then subjected to low-pass filtering with a cutoff frequency of 16 Hz. The signals were normalized and segmented into 32-s windows. SBP, DBP, and MAP values were calculated using the MIMIC III database’s every 4 s ABP signal window. Intermittent noninvasive blood pressure (NIBP) measurements were used to describe these values. Although the NIBP values were extracted every five minutes, the signals were sampled at 100 Hz. As a result, the missing values were substituted with the most recent NIBP values. Additionally, the time of this measurement was used as an input parameter. To compensate for the phase difference between the signals, cross-correlation was calculated between the ABP and PPG signals in the same manner as described previously [38]. Following that, the 32-s windows for identifying artifacts were chosen. Artifact-containing signals were removed from the final dataset. Numerous criteria were established to aid in the identification of the artifacts. Using a CNN network, the erroneous PPG signals were identified despite these criteria. Human identified signals were used to train the network. To obtain an accurate ABP signal, the pulse arrival time (PAT) was calculated using the ECG signal and the heart rate was calculated using the PPG signal.

In [75], the Fourier Transform (FFT) was used to remove unwanted information from both PPG and ABP signals. Then, for the PPG signal, a bandpass filter with cutoff frequencies of 0.1 Hz and 8 Hz was used, and for the ABP signal, a low-pass filter with a cutoff frequency of 5 Hz was used. Both signals have been normalized. Following that, each signal was divided into 256 samples with a 25% overlap.

#### 3.3.2. Data Availability

Deep learning algorithms need a huge amount of data for training, validation, and testing. Therefore, the availability of data is very important. Table 7 summarizes the use of data for different methods. The majority of papers make use of data from the MIMIC, MIMIC II, and MIMIC III waveform databases [76,77,78,79]. However, different papers used varying numbers of subjects, and the total amount of data collected varies as well. According to the total data, some papers divided the data into train, validation, and test sets, while others divided the data at the subject level. Several papers used the dataset for K-fold CV. For K-fold CV, each fold’s training data was partitioned into training and validation data. The number of patients, the total amount of data used in the algorithm, and the way the data is split all have a significant impact on the obtained results. The more diverse the data set, the more generalizable the proposed algorithm.

## 4. Result Comparison

All papers included figures to illustrate their estimated BP waveforms. The discussed works illustrated the obtained result by plotting the reference and estimated waveforms. Several papers used performance metrics to assess the obtained results, which are listed in Table 8. The majority of studies obtained SBP and DBP values from estimated waveforms. Additionally, some calculated the MAP values. Mean absolute error (MAE), mean error (ME), standard deviation (SD), root mean square error (RMSE), Pearson’s correlation coefficient (r), and average mean squared error (AMSE) are used as performance metrics. The majority of the papers also compared the obtained results to two standards: the British Hypertension Society (BHS) [80] and the Association for the Advancement of Medical Instrumentation (AAMI) [81,82]. The ME must be within ±5 mmHg, and SD must be less than or equal to 8 mmHg for data involving more than 85 subjects, according to the AAMI standard. Most of the papers defined the AAMI standard using the ME and some papers using the MAE [38,70,83]. In study [83], it is stated that calculating the mean error yields incorrect results because a lower ME may result in a higher MAE. We also found in the studies that some papers that showed lower ME [39,74] resulted in higher MAE. Therefore, in Table 8, the AAMI standard results on the basis of both ME and MAE are shown. Table 8 summarizes the performance metrics for estimation of BP waveforms, SBP, DBP, and MAP. Additionally, it illustrates the outcomes of the proposed works in terms of standards.

Ref. [34] illustrates the shapes of the BP waveforms at various body locations (neck, arm, radial artery, and dorsalis pedis of the foot). They compared the BP waveform obtained by their ultrasound probe to that obtained by the commercially available FDA-approved SphygmoCor EM3^®®^ device. As shown in Table 8, the work demonstrated ME for SBP and DBP. Using signal diagrams, the pressure-based methods [32,33] demonstrated the comparison of BP waveforms. No measurement metrics were used to compare the estimated waveforms’ accuracy.

Ref. [70] demonstrated the mean and average mean squared error for the estimated waveforms when compared to the waveforms in the MIMIC database, which are 3.4094 mmHg and 4.4797 mmHg, respectively. MAE and SD are within the range specified by the AAMI standard. Ref. [72] displays the RMSE, MAE, and mean square error for the estimated waveforms. For SBP and DBP, only the RMSE is displayed. In [73], BP waveforms can be estimated solely through the ECG, solely through the PPG, or both. The results are compared to pulse arrival time (PAT) models. However, no precise values for performance metrics are provided. The MAE of this BP waveform is 4.604 ± 5.043 mmHg in [37]. The AAMI criteria are met in the case of MAP and DBP but not in the case of SBP. The paper earned an A on the BHS standard. The GDCAE method of [40] was used to generate the results in Table 8 because it produces the best results among the three proposed methods of the paper. Despite having high performance metrics, this work does not meet the AAMI standard due to the small number of subjects. Ref. [38] compared the estimated waveform figures to the average Person’s correlation coefficient (r) value for BP waveforms. This work met both standards. In [39], the predicted ABP waveform (on the test set) was compared to the ground-truth ABP waveform based on the model—RDAE: with and without calibration. Five distinct cases were presented to illustrate the BP waveform results. The cases are available with a clear dicrotic notch, without a clear dicrotic notch, or without a clear dicrotic notch. Hypotension, with a clearly abnormal cycle and a high degree of BP fluctuation, was also included. The calibrated result is superior to the uncalibrated result. Thus, Table 8 displays the best result that satisfies the AAMI error range for ME values and not for MAE values. It received a grade B for estimating SBP for the BHS standard. Ref. [74] illustrates the result of BP waveforms with an example of an estimated BP waveform. The obtained BP values do not conform to either of the standards. Ref. [25] presented findings from two distinct datasets. For waveforms, the RMSE and r values are displayed. SBP and DBP values are given as mean and SD and the work conforms to the AAMI standard according to ME [25]. Ref. [75] displayed no result for waveform; even the estimated waveforms are not displayed. The results for SBP and DBP are presented, and the results satisfy the standards when the given value is used.

The results give a clear indication that no paper has given a well-established method for evaluating the waveforms’ obtained results. Different works employed a variety of performance metrics; in some cases, no metrics were used to demonstrate the accuracy of their conclusions. The majority of papers did not explain how to obtain the performance metrics for BP waveforms. As a result, it is exceedingly difficult to compare works solely on the basis of their outcomes.

## 5. Commercialization

Studies are being conducted to find some reliable and flexible BP sensors to measure BP noninvasively and to develop commercial devices [66,84,85,86]. Although several methods have been proposed to replace the invasive BP waveform measurement process, no reliable commercial device is yet available. In spite of the limitations, some pressure-based devices are proposed for commercial use as they can provide noninvasive BP waveform that is similar to the invasive one. The AtCor Sphygmocor Xcel device [87] and NIBP100A (Vasotrac) [88] are arterial tonometry-based devices. Under expert supervision, the AtCor Sphygmocor Xcel device can perform applanation tonometry on the radial artery and derive BP waveforms [34,68]. While the NIBP100A (Vasotrac) is capable of measuring BP waveforms without supervision, it is unable to measure beat-to-beat BP values [89]. The Vasotrac device proposes an alternate way to cuff calibration which provides the SBP, DBP, and MAP values every 12 to 15 pulses [90,91]. Finapres^®^ NOVA [92,93,94] and CNAP^TM^ Monitor 500 [32,95,96,97] are additional devices that operate on the principle of vascular unloading. Another device, CNAP2GO finger-ring [98], is currently conducting research to commercialize it. Ref. [85] proposed a flexible PDMS-DI water dielectric sensor and applied the sensor in OMRON and Fluke NIBP analyzer to obtain BP values. The NIBP analyzer was used to obtain the oscillometric waveform of the SBP and DBP values. However, the flexible sensor is not applied in devices that can measure complete BP waveforms. All devices are based on pressure, which has a number of drawbacks as listed in Table 1 and Table 2. There is currently no commercially available ultrasound or deep-learning-based device.

## 6. Discussion and Future Prospects

The purpose of this review article is to summarize the research on BP waveform estimation and to provide an overview of BP waveform measurement approaches. Three noninvasive BP waveform monitoring techniques have been introduced thus far. However, each of these methods has some limitations, and their use in hospitals and on a routine basis is not yet confirmed. From a pathophysiological standpoint, the BP waveform is a significant and direct predictor of the majority of ischemic heart diseases [99]. Continuous and long-term monitoring of such occurrences can lead to significant breakthroughs in cardiac disease prevention and detection, which are presently unachievable with current medical instruments [5,100,101].

Development of the methods in a chronological order over the years to predict the BP waveform is shown in Figure 6. Figure 6 displays that ultrasound and deep learning are the latest methods in this sector. However, research is still going on in every noninvasive method. PPG signals are used in one of the pressure-based techniques (vascular unloading) and in the majority of machine-learning-based techniques. Due to the fact that these methods require biosignals, distinct disadvantages associated with biosignals persist for these methods. The penetration depth of PPG signals, in particular, is insufficient for assessing the central vasculature [102]. Other technical issues with PPG signals include signal aliasing due to arterial and venous palpitations [103], vulnerability to body moisture and high temperature [104], and a strong reliance on the composition of blood staying consistent [105]. Additionally, the PPG signal is susceptible to a variety of artifacts. Skin tone also plays a role in the measurement of PPG signals [106]. Several studies used additional biological signals, such as the electrocardiogram, to obtain accurate measurement waveforms. However, it cannot compete with the use of PPG signals. Thus, in some cases, ECG signals have been used to improve the predicted BP waveforms based on PPG signals. Another pressure-based technique known as tonometry, as well as an ultrasound-based technique, do not require biosignals. However, only two studies utilizing ultrasound were discovered, and they also have some significant limitations. Although tonometry has been extensively studied for a long period of time, it cannot yet fully replace invasive measurement methods due to inherent limitations.

Our work was also to survey the performance metrics used to evaluate the proposed methods. As illustrated in Table 8, not all studies use the same performance metrics. There are no definitive metrics for the waveform estimation result. The majority of papers did not employ any metrics to evaluate the estimated waveforms [32,33,34,39,73,74,75]. The accuracy of waveforms is demonstrated by plotting several waveforms against their reference waveforms. One or two graphs are used to illustrate this. However, in the absence of specific performance metrics, it is difficult to rely on the examples, as the majority of papers demonstrated the best–estimated waveforms. Numerous papers demonstrated the accuracy by estimating the commonly used SBP and DBP values from the predicted waveforms. Additionally, some papers estimated MAP values. Table 8 summarizes the most frequently used performance metrics. Although some work reports excellent performance, it is difficult to compare it to other studies due to the disparity in reported metrics and the data used to replicate the proposed method.

Due to the dataset’s lack of similarity, it is difficult to compare the results obtained using various methods. The pressure-based and ultrasound-based methods estimated the BP waveforms using mathematical equations. They did not make a point of emphasizing the results obtained from various subjects. As a result, the studies either used a customized dataset or evaluated the process with a small sample size. On the other hand, methods based on machine learning require a large amount of data to build their model. While there is a wealth of data on the use of arterial SBP and DBP and their medical and diagnostic utility, there are relatively few datasets of BP waveforms. Nonetheless, different versions of the MIMIC dataset are used for machine learning methods because this is the only publicly available dataset that contains invasive BP waveforms, also known as ABP waveforms, with respect to the PPG signal, ECG signal, and cuff-based noninvasive measurement values (SBP and DBP). However, these datasets contain information on over 40,000 subjects. The number of subjects, the duration of the data, and the type of data vary according to the research work, as indicated in Table 7. Even when using similar datasets, it is impossible to compare the results obtained by different authors due to this variability. As a result, the data must be collected and preprocessed according to a documented protocol. Following that, you can select from a variety of deep learning models. Because the studies on machine learning included a variety of deep learning algorithms, there is no clear preference for one technique over another. However, recent work has concentrated on encoder–decoder models with the primary objective of estimating the BP waveforms as stated in Table 4. The selected deep learning algorithm largely depends on the type of data used to build the algorithm.

However, one disadvantage of deep learning is the training time, which is dependent on the network architecture and the hardware used to train the model, both of which are costly. Training time can be halved by utilizing appropriate hardware to train the networks [107]. Typically, the use of a GPU resolves this issue [108]. However, it is critical to make the best use of GPU resources when training deep learning networks and performing a large number of floating-point calculations and matrix computations with greater efficiency than CPUs [109]. Some software or development environments, such as Python and MATLAB, are utilized for this reason. The key benefit of these programming languages is that they are written in such a way that the creation of deep learning algorithms becomes easier without requiring a thorough understanding of GPU hardware structure. Nonetheless, if computational materials are required to be purchased or acquired, these algorithms could be costly to implement. This issue was discussed in [39], and a deep learning model with a comparatively small number of parameters was proposed.

Future research in this field has enormous potential. The datasets and records that were analyzed can be made publicly accessible. Due to the fact that PPG contains a variety of artifacts and that reliable accusation is dependent on a variety of factors, additional biological parameters such as the subject’s gender, age, BMI, and habits can be used to obtain more accurate results. A universal performance metric or graph must be established to allow for proper comparisons between methodologies. The deployment of deep learning models for BP waveform prediction in hardware could be investigated further. The method for predicting BP waveforms in any situation must be studied for generalizability. None of the methods examined can provide generalizable results for both normal and intensive care unit patients. After obtaining the result, calibration with noninvasive cuff-based values or a reference invasive waveform is required. The reliance on cuff-based or invasive measurements should be eliminated. In general, a long-term study is necessary to establish a promising result for predicting BP waveforms and implementing the system in hospitals and homes.

## 7. Conclusions

This work provided an overview of the method for estimating BP waveforms as well as the significance of the waveforms. Our findings indicate that three distinct types of studies are currently being conducted to estimate BP waveforms. They are based on methods such as pressure, ultrasound, and machine learning. Numerous biosignals, particularly the use of PPG signals, are observed to be the most useful in predicting these useful waveforms. Due to the fact that these biosignals may contain a variety of artifacts, it is critical to conduct research on them to ensure that biosignals are obtained without error and that accurate results are obtained. BP waveforms contain a wealth of information that can be used to aid in the treatment of cardiovascular diseases. As a result, it is crucial that the obtained results are accurate. Our review concluded that each study conducted to date has some flaws. However, there are numerous research opportunities in this field that could result in the noninvasive and accurate acquisition of BP waveforms.

The field of noninvasive BP waveform estimation is a promising yet challenging one. In future work, a better understanding of ultrasound, different biosignal information, and machine learning should enable researchers to address the aforementioned issues and ensure the successful development of technologies for noninvasive BP waveform estimation. The incorporation of these methods for noninvasively estimating BP waveforms, which may indicate probable cardiac failure in the event of operational blood loss, represents the evolution of invasive BP measurement.

## Figures and Tables

**Figure 1 sensors-22-03953-f001:**
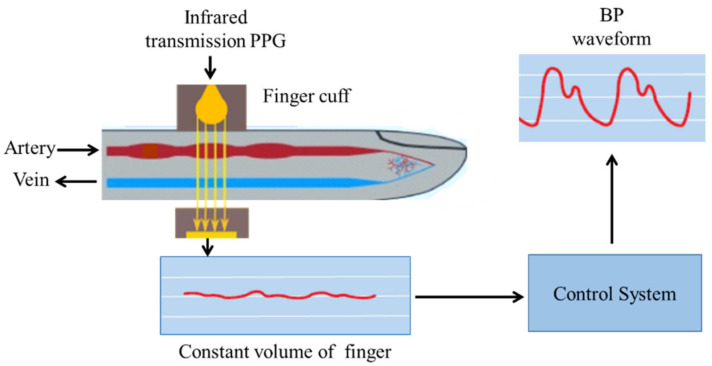
Noninvasive BP waveform estimation using the vascular unloading technique.

**Figure 2 sensors-22-03953-f002:**
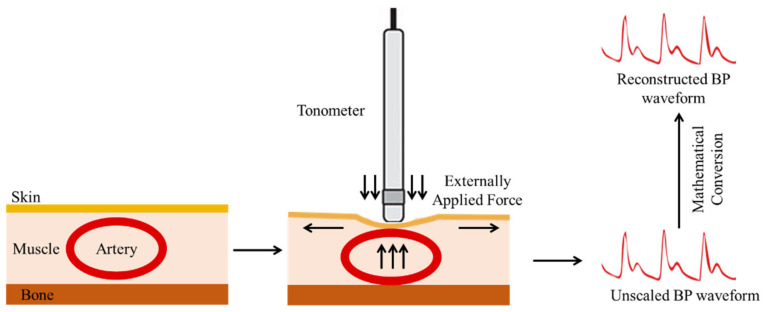
Noninvasive BP waveform estimation using the arterial tonometry method.

**Figure 3 sensors-22-03953-f003:**
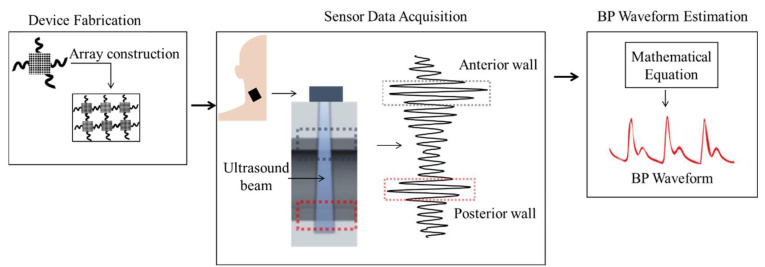
Noninvasive BP waveform estimation using a wearable ultrasound probe.

**Figure 4 sensors-22-03953-f004:**
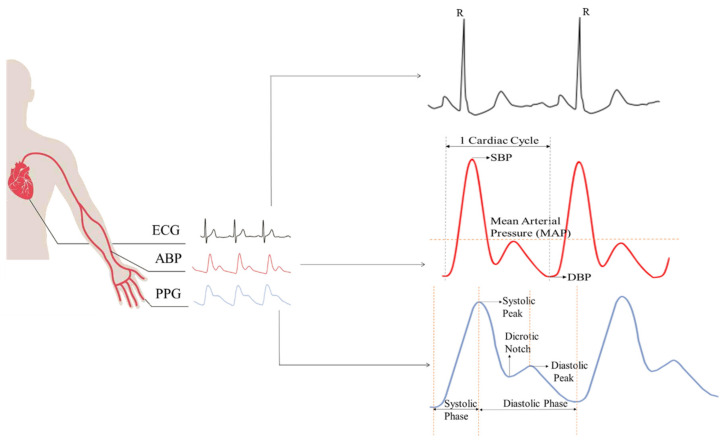
Photoplethysmogram (PPG), electrocardiogram (ECG), and arterial blood pressure (ABP) signals with their measuring positions and characteristics.

**Figure 5 sensors-22-03953-f005:**
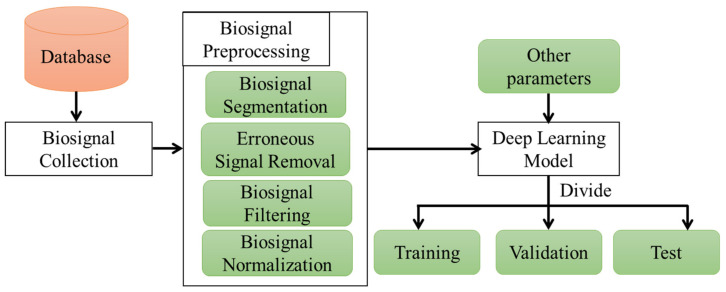
Noninvasive BP waveform estimation using biosignals and deep learning algorithms.

**Figure 6 sensors-22-03953-f006:**
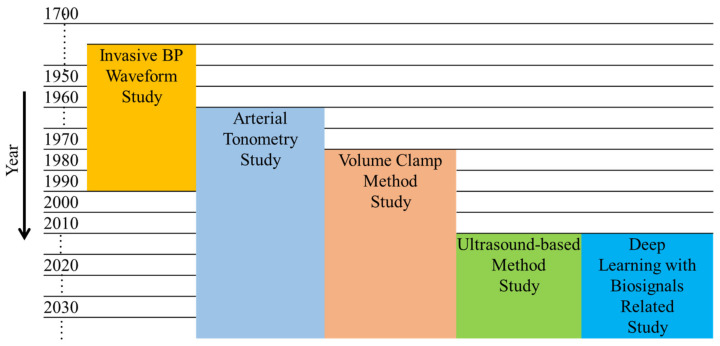
Development of different methods over the years. Invasive blood pressure waveform study [17,18,19,20,21,22,23]; Arterial tonometry study [33,65,67,68,86,89,90,91]; Volume clamp method study [32,55,56,57,58,59,60,61,62,63,64,92,93,94,95,96,97]; Ultrasound-based method study [34,35,36]; Deep learning with biosignals related study [25,37,38,39,40,70,72,73,74,75].

**Table 1 sensors-22-03953-t001:** Advantages and disadvantages of the vascular unloading technique to estimate BP waveforms.

Advantages	Disadvantages
A noninvasive continuous monitoring of BP waveform is possible. No risk of infection like the invasive method. Commercial devices are available.	Wearing a cuff for an extended period of time is uncomfortable and causes numbness and arterial congestion in the measurement finger. Different finger-widths require different cuff sizes. The finger’s thin arteries are responsible for thermoregulation. As a result, they are susceptible to vasoconstriction and vasodilation in response to both external temperature and the individual’s volume status. There is no guarantee that the finger arteries’ pressure will be comparable to that of the large arteries. This method necessitates the use of PPG signals, which present unique technical difficulties.

**Table 2 sensors-22-03953-t002:** Advantages and disadvantages of the arterial tonometry to estimate BP waveforms.

Advantages	Disadvantages
Do not need finger cuff. A noninvasive continuous monitoring of BP waveform is possible. No risk of infection like the invasive method. Less sensitive to diseases such as vasoconstrictions caused by using finger cuffs. Commercial devices are available.	Used only when a bony system is available to provide firm mechanical support [65]. Ineffective approach if the person is obese, as the propagation of pulse waves to the skin is substantially slowed. The accurate placement of the measurement device on the middle of the artery is very critical. A commercial BP cuff device is required for calibration. There should be no movement during the measurement process [34].

**Table 3 sensors-22-03953-t003:** Advantages and disadvantages of the ultrasound-based method to estimate BP waveforms.

Advantages	Disadvantages
The wearable ultrasound probe is stretchable. The method is resistive to motion artifacts. No risk of infection like the invasive method. Can measure BP waveforms for a long time. The probe can be used on different measurement sites (radial artery, carotid artery, brachial artery, pedal artery).	Calibration is required, and the calibration coefficient is dependent on both the DBP and the vessel rigidity coefficient. Calibration is required prior to and following any physiological change, such as that caused by exercise. A BP cuff is required to obtain the DBP. The device is not tested on a variety of subjects because the coefficients will vary according to the subject. No commercial device is available yet.

**Table 4 sensors-22-03953-t004:** Summary of the deep learning–based methods and input biosignals to estimate BP waveforms.

Authors	Pub. Year	Method	Input	Input Length
[70]	2015	Wavelet neural network	PPG	Not given
[72]	2016	Long Short-Term Memory (LSTM)	PPG	Not specific
[73]	2020	Nonlinear autoregressive models with exogenous input (NARX) with ANN	ECG or PPG or both, two BP data	100 samples
[37]	2020 (preprint server)	U-Net and 1D MultiResUNet	PPG	8 s
[40]	2020	Deep convolutional autoencoder (DCAE)	PPG	5 s
[38]	2021	1D U-Net	PPG	256 samples = 2.048 s with overlapping
[39]	2021	Regularized deep autoencoder (RDAE)	PPG	625 samples = 5 s
[74]	2021 (preprint server)	U-Net	PPG	32 samples
[25]	2021	1D V-Net	ECG, PPG, most recent cuff-based SBP, DBP, and MAP values, the time of these values, the standard deviation and median of the pulse arrival time, and pulse rate	4 s
[75]	2022 (preprint server)	Cycle generative adversarial network (CycleGAN)	PPG	256 samples = 2.048 s with overlapping

**Table 5 sensors-22-03953-t005:** Advantages and disadvantages of deep learning–based methods to estimate BP waveforms.

Ref.	Advantages	Disadvantages
[70]	Only PPG signal is needed. An optimized neural network has been proposed.	High computational complexity. Data redundancy.
[72]	Only PPG signal is needed.	Length of input is not defined. The network architecture and overall process are not described. Only one point output w.r.t multiple input point.
[73]	A feedback loop is used to predict BP values BP waveform estimation is shown from ECG signal which is less sensitive to artifacts.	BP data is needed for model training. Few subjects. Delay removal process is not applicable for all cases. Different ECG, PPG, and BP peak ranges were not unified while training. cCoss-correlation analysis was performed to quantify any delay between the predicted and the measured BP. Two BP waveform points are needed for the input of the ANN.
[37]	Only PPG signal is needed. 10-fold cross–validation is done with the data.	One deep learning network is needed for approximation, and another deep learning network is needed for estimating the accurate waveform.
[40]	Only PPG signal is needed. Custom data has been used. The number of subjects is less.	GDCAE method ensembles two deep learning algorithms to get accurate SBP and DBP values.
[38]	Only PPG signal is needed. Comparatively good result is obtained using only one model.	Same subjects are used for training and testing.
[39]	Only PPG signal is needed. The proposed model requires fewer parameters than other methods. Subjects of training and testing sets are different.	The calibrated model gives better result.
[74]	The model is implemented on a Raspberry Pi 4 device. Only PPG signal is needed.	The device implementation process is not described. PPG signal artifact can provide wrong results.
[25]	SBP and DBP estimation process is shown. Results were shown for two different datasets. Training and validation sets include different patients.	Both the PPG and ECG waveforms and several constants are needed as input.
[75]	Only PPG signal is needed. 5-fold cross–validation is obtained with the data.	PPG signal artifact can provide wrong results. Constant value of λ is used which is set to 10.

**Table 6 sensors-22-03953-t006:** Different data preprocessing techniques of deep learning methods. In the normalization equations, $x^{\prime }$ is the normalized signal window, x denotes the signal window before normalization, μ is the mean, and σ is the standard deviation.

Ref.	Preprocessing Steps	Normalization Equation
[70,72,74]	Filtering the biosignals Removing erroneous biosignals	-
[40,73]	Removing erroneous biosignals	-
[37]	Filtering the biosignals Removing erroneous biosignals Normalization	$x^{\prime }=\frac{x-\mu }{\max\left(x\right)-\min\left(x\right)}$
[25,38]	Filtering the biosignals Segmentation Removing erroneous biosignals Normalization	For [38], $x^{\prime }=\frac{x-\min\left(x\right)}{\max\left(x\right)-\min\left(x\right)}$ For [25], $x^{\prime }=\left(x-\min\left(x\right)\right)\times \frac{SBP-\mathrm{DBP}}{\max\left(x\right)-\min\left(x\right)}+DBP$
[39]	Filtering the biosignals Removing the erroneous portion of biosignals Segmentation Normalization	$x^{\prime }=\left(\frac{x-\mu }{\sigma }\right)$
[75]	Removing the erroneous portion of biosignals Filtering the biosignals Segmentation	-

**Table 7 sensors-22-03953-t007:** Summary of the datasets used to train, validate, and test the deep learning models.

Ref	Dataset	# of Subject	Total Data (in hours)	K-Fold Cross-Validation	Train:Val:Test
[70]	MIMIC	>90	-	No	Not given
[72]	MIMIC	42	-	No	80:10:10 (in total data)
[73]	MIMIC II	15	-	No	70:15:15 (in total data)
[37]	MIMIC II	942	≈353.5	Yes (10 Folds)	78.58:-:21.42 (in total data)
[40]	Custom	18	≈50.72	Yes (10 Folds)	85:-:15 (in total data)
[38]	MIMIC, MIMIC III Waveform	100	≈195	No	70:15:15 (in total data)
[39]	MIMIC II	1227	≈54.53	No	60:20:20 (in subjects)
[74]	MIMIC II Waveform database	948	≈353.5	Yes (10 Folds)	78.58:-:21.42
[25]	MIMIC III, UCLA	MIMIC-264, UCLA-110	≈2516.48	No	66:-:33 (in subjects of MIMIC)
[75]	MIMIC II Waveform database	92	≈7.67	Yes (5 Folds)	80:-:20

**Table 8 sensors-22-03953-t008:** Performance summary of all the discussed BP waveform estimation methods. “--” is used where metric is not used.

Method	Ref.	Year	Performance Metrics (no Unit for r, mmHg for Others)				BHS Grade	AAMI
			Waveform	SBP	DBP	MAP		
Ultrasound-Based	[34]	2018	-	ME: 0.05	ME: 0.28	-	-	-
Pressure-Based	[32]	2006	-	-	-	-	-	-
	[33]	2015	-	-	-	-	-	-
Deep Learning-Based	[70]	2015	Mean: 3.4094 AMSE: 4.4797	MAE±SD: 2.32 ± 2.91	MAE±SD: 1.92 ± 2.47	-	-	Passed (MAE)
	[72]	2016	RMSE: 6.042 ± 3.26 r: 0.95 MAE: 5.98 ME: −0.214	RMSE: 2.575	RMSE: 1.977	-	-	-
	[73]	2020	-	-	-	-	-	Failed
	[37]	2020 (preprint server)	MAE±SD: 4.604 ± 5.043	MAE±SD: 5.727 ± 9.162	MAE±SD: 3.449 ± 6.147	MAE±SD: 2.310 ± 4.437	A	Failed
	[40]	2020	RMSE: 3.46 MAE: 2.33 r: 0.984	RMSE: 3.41 MAE: 2.54 r: 0.981	RMSE: 2.14 MAE: 1.48 r: 0.979	-	-	Failed (subjects < 85)
	[38]	2021	r: 0.993	MAE±SD: 3.68 ± 4.42 RMSE: 5.75 r: 0.976	MAE±SD:1.97 ± 2.92 RMSE: 3.52 r: 0.970	MAE±SD: 2.17 ± 3.06 RMSE: 3.75 r: 0.976	A	Passed (MAE)
	[39]	2021	-	ME±SD: 1.648 ± 6.640 MAE: 5.424	ME±SD: 1.280 ± 3.740 MAE: 3.144	ME±SD: −0.304 ± 3.412 MAE: 2.885	SBP:B	Passed (ME) Failed (MAE)
	[74]	2021 (preprint server)	-	ME±SD: −0.225 ± 8.504 MAE: 5.16	ME±SD: 0.594 ± 4.778 MAE: 2.89	ME±SD: 0.425 ± 4.784	SBP:B	Passed (ME) Failed (MAE)
	[25]	2021	MIMIC RMSE: 5.823 MIMIC r: 0.957 UCLA RMSE: 6.961 UCLA r: 0.947	MIMIC ME±SD: 4.297 ± 6.527 UCLA ME±SD:2.398 ± 5.623	MIMIC ME±SD:−3.114 ± 4.570 mmHg UCLA ME±SD: −2.497 ± 3.785	-	-	Passed (ME)
	[75]	2022 (preprint server)	-	MAE±SD: 2.89 ± 4.52 RMSE: 5.18 ME: 0.67 r: 0.97	MAE±SD:3.22 ± 4.67 RMSE: 4.82 ME: 1.78 r: 0.94	-	A	Passed (MAE, ME)

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
