# Peer review of "A Review of Noninvasive Methodologies to Estimate the Blood Pressure Waveform"

_sensors, 2022, doi:10.3390/s22103953_

Round 1
Reviewer 1 Report
I have the following suggestions:
- The main title and emphasis of the review paper is to summarize all the non-invasive approaches to estimating the BP waveforms. Then why “non-invasive BP monitoring” is not searched as a keyword? There are several recent papers and reviews on non-invasive wearable devices are published that cover several devices for diverse applications (https://doi.org/10.3390/electronics11050716, https://doi.org/10.3390/s21134273). These papers should be considered while working on the non-invasive BP monitoring techniques.
- The author said, “However, to our knowledge, this is the first paper to review 113 non-invasive methods for estimating BP waveforms.” I think the author should be a little careful with such bold remarks, there are several review papers available on such topics such as https://doi.org/10.3389/fmed.2017.00231, https://doi.org/10.1016/j.irbm.2014.07.002.
- I am not sure if the author has self-drawn the figures or taken them from some source. If it’s taken from the literature, then the figures should be properly cited. I don’t see a citation to the original source in the figures’ caption.
- The paper is written in the form of a book chapter where the main techniques are explained in detail which is not desired. It can be found in the literature. However, the main aim of a review paper is to collect the previously published paper on a certain topic and explain them with graphical illustrations. In this review, there is a lack of graphical images of non-invasive devices working on different mechanisms. Therefore, I suggest the author add more figures from authentic sources and discuss their main attributes and make a comparison to ensure which technique is better/accurate and why.
- The author said in section 6, “Although some work reports excellent performance,” “The majority of papers did not employ any metrics to evaluate the estimated waveforms,” Please give references when using such statements. Otherwise, it is not clear which work the author is referring to.
Author Response
Thank you for your valuable reviews. The response to the reviews is stated in detail in the attached document.

Reviewer 2 Report
Accurate estimation of blood pressure waveforms is critical for ensuring the safety and proper care of patients in ICUs or during operation. In the manuscript of sensors-1691374, the authors try to summarize the current state of knowledge regarding the BP waveform, three methodologies
(pressure-based, ultrasound-based, and deep learning-based) used in non-invasive BP waveform. The authors claimed that it is the first paper to review non-invasive methods for estimating BP waveforms, but the following references (not included in their manuscript) did not support the conclusions.
1. Cuff-Less and Continuous Blood Pressure Monitoring: A Methodological Review; Technologies 2017, 5, 21; doi:10.3390/technologies5020021
2. Recent Research and Developing Trends of Wearable Sensors for Detecting Blood Pressure; Sensors 2018, 18, 2772; doi:10.3390/s18092772
3. Advances in Non-Invasive Blood Pressure Monitoring
Sensors 2021, 21, 4273. https://doi.org/10.3390/s21134273
4. Blood Pressure Sensors: Materials, Fabrication Methods, Performance Evaluations and Future
Perspectives
Sensors 2020, 20, 4484; doi:10.3390/s20164484
5. Cuffless Blood Pressure Measurement; Annu. Rev. Biomed. Eng. 2022. 24:201–228, https://doi.org/10.1146/annurev-bioeng-110220-014644
6. Cuffless Blood Pressure Monitoring: Promises and Challenges; DOI: 10.2215/CJN.03680320
7. Techniques for Non-Invasive Monitoring of Arterial Blood Pressure; DOI: 10.3389/fmed.2017.00231
8. Non-invasive continuous blood pressure monitoring: a review of current applications; Front. Med. 2013, 7(1): 91–101; DOI: 10.1007/s11684-013-0239-5
Author Response

(The authors gave the same response as above.)

Reviewer 3 Report
This paper has summarized three current methodologies used for noninvasive BP waveform estimation, including pressure-based, ultrasound-based, and deep learning-based; and discussed the shortcomings of these methods as well as future directions. Overall, the paper is well structured and the literature cited is adequate.
My comments on this paper are:
first, for pressure-based methods, there are also solutions in the literature that use flexible pressure sensors to detect BP waveforms, and this part of the description needs to be added;
second, it would be better to add a diagram to visualize the development of each method (in chronological order) or the advantages and disadvantages of each;
finally, there may be an error in line 458, where AAMI requires ME less than 5 mmHg, not MAE.
Author Response
Thank you for your valuable reviews. The response to the reviews is stated in detail in the attached document. Please see the attachment.

Round 2
Reviewer 1 Report
The authors have revised the paper as per the reviewer's suggestions. I am willing to accept the paper in its current form.
Author Response
Thank you so much for accepting our paper.
Reviewer 2 Report
The authors have revised and improved the paper as the reviewer's suggestions, so I am willing to suggest the paper to be accepted in its current form.
Author Response

(The authors gave the same response as above.)

Reviewer 3 Report
Thank you for revising the manuscript according to my previous comments. I have just one question. According to the ANSI/AAMI/ISO [1], “For systolic and diastolic blood pressure, then mean value of the differences of the determinations, … for all subjects shall be within or equal to ±5 mmHg …”, therefore, AAMI requires ME (mean error or mean difference) instead of MAE.
[1] Association for the Advancement of Medical Instrumentation. Non-invasive sphygmomanometers–Part 2: clinical investigation of automated measurement type ANSI/AAMI. ISO 81060-2/ANSI-AAMI, 2nd ed. Arlington, VA: AAMI, 2013.
